# Accurately Solving Rod Dynamics with Graph Learning

**Han Shao**
KAUST
han.shao@kaust.edu.sa

**Tassilo Kugelstadt**
RWTH Aachen University
kugelstadt@cs.rwth-aachen.de

**Torsten Hädrich**
KAUST
torsten.hadrich@kaust.edu.sa

**Wojciech Pałubicki**
AMU
wp06@amu.edu.pl

**Jan Bender**
RWTH Aachen University
bender@cs.rwth-aachen.de

**Sören Pirk**
Google Research
pirk@google.com

**Dominik L. Michels**
KAUST
dominik.michels@kaust.edu.sa

## Abstract

Iterative solvers are widely used to accurately simulate physical systems. These solvers require initial guesses to generate a sequence of improving approximate solutions. In this contribution, we introduce a novel method to accelerate iterative solvers for rod dynamics with graph networks (GNs) by predicting the initial guesses to reduce the number of iterations. Unlike existing methods that aim to learn physical systems in an end-to-end manner, our approach guarantees long-term stability and therefore leads to more accurate solutions. Furthermore, our method improves the run time performance of traditional iterative solvers for rod dynamics. To explore our method we make use of position-based dynamics (PBD) as a common solver for physical systems and evaluate it by simulating the dynamics of elastic rods. Our approach is able to generalize across different initial conditions, discretizations, and realistic material properties. We demonstrate that it also performs well when taking discontinuous effects into account such as collisions between individual rods. Finally, to illustrate the scalability of our approach, we simulate complex 3D tree models composed of over a thousand individual branch segments swaying in wind fields.

## 1 Introduction

The numeric simulation of a dynamic system commonly comprises the derivation of the mathematical model given by the underlying differential equations and their integration forward in time. In the context of physics-based systems, the mathematical model is usually based on first principles and depending on the properties of the simulated system, the numerical integration of a complex system can be very resource demanding (Nealen et al., 2006), e.g., hindering interactive applications. Enabled by the success of deep neural networks to serve as effective function approximators, researchers recently started investigating the applicability of neural networks for simulating dynamic systems. While many physical phenomena can well be described within fixed spatial domains (e.g., in fluid dynamics) that can be learned with convolutional neural network (CNN) architectures (Chu & Thuerey, 2017; Guo et al., 2016; Tompson et al., 2016; Xiao et al., 2020), a large range of physical systems can more naturally be represented as graphs. Examples include systems based on connected particles (Müller et al., 2007), coupled oscillators (Michels & Desbrun, 2015; Michels et al., 2014),

35th Conference on Neural Information Processing Systems (NeurIPS 2021).

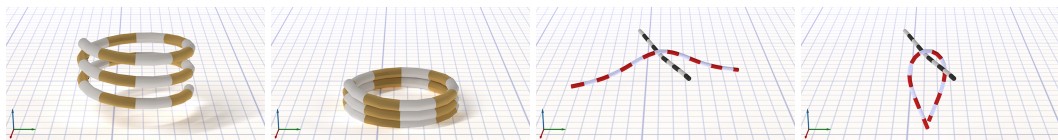

Figure 1: Renderings taken from real-time simulations of the elastic deformation of a helix falling down on the ground plane (left) and two rods colliding with each other (right).

or finite elements (Nealen et al., 2006). Existing methods enable learning these systems often in an end-to-end manner and with a focus on replacing the entire or a part of the integration procedure. A number of methods show initial success in approximating physical systems; however, they often fail to reliably simulate the state of a system over longer time horizons if significant disadvantages are not accepted, such as the use of large datasets containing long-term simulations and the employment of specific memory structures (Sanchez-Gonzalez et al., 2020).

In this paper, we aim to improve the performance of iterative solvers for physical systems with graph networks (GN). An iterative solver requires an initial guess, and based on it generates a sequence of improving approximate solutions. The initial guess can be computed by simply using values obtained in the previous iteration or by solving a few steps of a similar but simpler physical system. The performance of an iterative solver is significantly influenced by the calculation of the initial guess, which we aim to replace with the prediction of a GN. To demonstrate our approach, we use a position-based dynamics (PBD) solver that approximates physical phenomena by using sets of connected vertices (Bender et al., 2017, 2014b; Macklin et al., 2016; Müller et al., 2007). To simulate a physical system, PBD first computes updated locations of vertices using symplectic Euler integration to then correct the initial position estimates so as to satisfy a set of predefined constraints. The correction step is known as *constraint projection* and is commonly solved iteratively. The explicit forward integration for predicting the system's updated state has negligible cost, whereas the projection step is computationally expensive. Our goal is to employ a GN to predict the outcome of the constraint projection step as an initial guess. This way, our approach inherits the long-term stability of a classic PBD solver, while providing better run-time performance.

To showcase the capabilities of our combined PBD solver, we aim to simulate the physically plausible mechanics of elastic rods. Rods play an important role for a variety of application domains, ranging from surgical simulation of sutures (Feess et al., 2016), catheters, and tendons (Pai, 2002), to human hair (Michels et al., 2015) and vegetation (Pirk et al., 2017) in animated movies. Furthermore, approaches exist to realistically simulate rods as sets of connected vertices accurately capturing their mechanical properties (Bergou et al., 2008; Kugelstadt & Schoemer, 2016; Michels et al., 2015; Pai, 2002). Our approach is able to generalize across different initial conditions, rod discretizations, and realistic material parameters such as Young's modulus and torsional modulus (Deul et al., 2018). Moreover, we demonstrate that our approach can handle discontinuous collisions between individual rods. Figure 1 shows examples of elastic rod deformations of a helix falling down (left) and two colliding rods (right). Finally, we show that the data-driven prediction of the initial guesses of the constraint projection leads to a decreased number of required iterations, which – in turn – results in a significant increase of performance compared to canonical initial guesses.

In summary, our contributions are: (1) we show how to accelerate iterative solvers with GNs; (2) we show that our network-enabled solver ensures long-term stability required for simulating physical systems; (3) we showcase the effectiveness of our method by realistically simulating elastic rods; (4) we demonstrate accuracy and generalizability of our approach by simulating different scenarios and various mechanical properties of rods including collisions and complex topologies (dynamic tree simulations).

## 2    Related Work

In the following we provide an overview of the related work, which spans from data-driven physics simulations and graph learning to position-based dynamics and elastic rods.

**Data-driven Physics Simulations.** It has been recognized that neural networks can be used as effective function approximators for physical and dynamic systems. To this end, early approaches focus on emulating the dynamics of physics through learned controllers (Grzeszczuk et al., 1998) or by designing subspace integrators (Barbič & James, 2005). Today, a range of approaches exist that enable learning ordinary and partial differential equations (Lagaris et al., 1998; Raissi et al., 2019; Raissi & Karniadakis, 2018), for example, to transform them into optimization problems (Dissanayake & Phan-Thien, 1994), to accelerate their computation (Mishra, 2018; Sirignano & Spiliopoulos, 2018), or to solve for advection and diffusion in complex geometries (Berg & Nyström, 2018). Other methods focus on specific data-driven solutions for non-linear elasticity (Ibañez et al., 2017), for approximating Maxwell's equation in photonic simulations (Trivedi et al., 2019), or for animating cloth (Wang et al., 2011), partially focusing on interactive applications (Holden et al., 2019). More recently, research on data-driven approaches for modeling the intricacies of fluid dynamics has gained momentum (Ladický et al., 2015; Ummenhofer et al., 2020). Due to fixed-size spatial representation of Eulerian fluid solvers, a number of approaches rely on CNN-type architectures (Chu & Thuerey, 2017; Guo et al., 2016; Tompson et al., 2016; Xiao et al., 2020). Furthermore, it has been shown that data-driven approaches can even be used to approximate the temporal evolution of fluid flows (Wiewel et al., 2018), to compute liquid splashing (Um et al., 2017), artistic style-transfer (Kim et al., 2020), or to derive fluid dynamics from reduced sets of parameters (Kim et al., 2019).

**Graph-based Learning.** Graphs have proven to be a powerful representation for learning a wide range of tasks (Battaglia et al., 2018; Scarselli et al., 2009). In particular, it has been shown that graphs enable learning knowledge representations (Kipf et al., 2018), message passing (Gilmer et al., 2017), or to encode long-range dependencies, e.g., as found in video processing (Wang et al., 2017). A variety of methods uses graph-based representations to learn properties of dynamic physical systems, e.g. for climate prediction (Seo & Liu, 2019), with an emphasis on individual objects (Chang et al., 2016) and their relations (Sanchez-Gonzalez et al., 2018), for partially observable systems (Li et al., 2018b), the prevalent interactions within physical systems (Kipf et al., 2018), hierarchically-organized particle systems (Mrowca et al., 2018), or – more generally – physical simulation (Sanchez-Gonzalez et al., 2019, 2020). While many of the existing approaches learn the time integration of physical systems in an end-to-end manner, we use a graph network to predict the outcome of a PBD solver for rod dynamics to enable more efficient computations.

**Position-based Dynamics and Elastic Rods.** PBD is a robust and efficient approach for simulating position changes of connected sets of vertices (Bender et al., 2017, 2014b; Macklin et al., 2016; Müller et al., 2007). Compared to forced-based methods that compute the force directly, the interaction between different vertices in PBD is realized by a constraint projection step in an iterative manner. To avoid the dependency of the system's stiffness on the number of iterations and the time step size, an extended position-based dynamics approach was introduced (XPBD) (Macklin et al., 2016). A number of methods exist that model the dynamic properties of rods that can even simulate more complicated rod mechanics (Pai, 2002). Moreover, particle systems were employed to simulate the dynamics of rods (Michels et al., 2017) and, in particular, for the physically accurate simulation of thin fibers (Michels et al., 2015) such as present in human hair or textiles. On a different trajectory, it has been recognized that rods can be simulated based on PBD (Umetani et al., 2014). The initial formulation was improved (Kugelstadt & Schoemer, 2016) by including the orientation of rod segments in the system's state to account for torsion effects. Later, the XPBD framework was utilized (Deul et al., 2018) to address the non-physical influence of iteration numbers and steps sizes, which enables the more accurate simulation of elastic rods.

## 3 Methodology

We propose a novel approach to simulate the temporal evolution of a dynamic system which consists of elastic rods. Each rod is discretized using several rod segments arranged along its centerline (Figure 2). For each rod segment, its state is described by its position, orientation, velocity and angular velocity. The state of the system is given as the set of the individual states of all rod segments. The simulation is carried out by employing PBD (Müller et al., 2007) directly manipulating the system's state. Orientations are represented as quaternions allowing for a convenient implementation of bending and twisting effects (Kugelstadt & Schoemer, 2016). Extended PBD (XPBD) (Macklin et al., 2016) is implemented to avoid that the rods' stiffnesses depends on the time step size and the

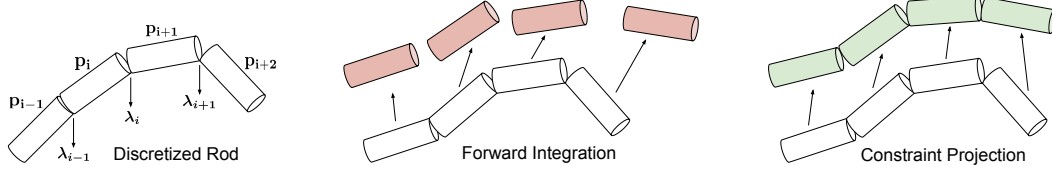

Figure 2: Illustration of the discretization of a single rod using several rod segments arranged along its centerline (left). Each rod segment is described by its position and orientation within the generalized coordinates $\mathbf{p}_i$. The Lagrange multipliers $\boldsymbol{\lambda}_i$ represent the interaction between rod segments. The forward integration path is illustrated in red (middle) and constraint projection in green (right).

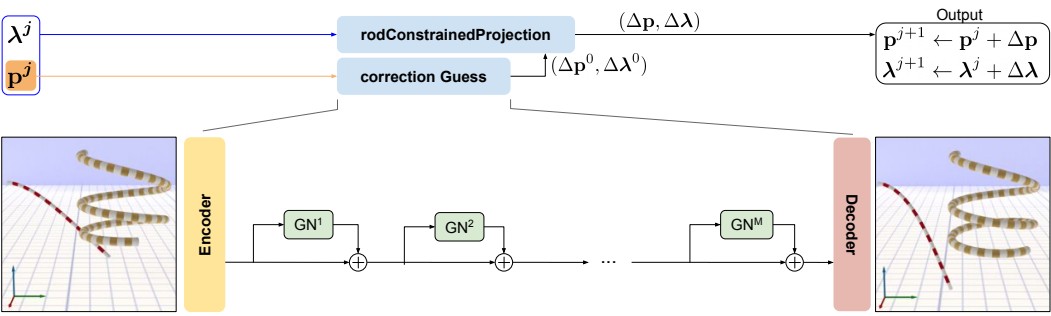

Figure 3: Illustration of our approach incorporating a network which consists of $M$ graph networks (GN-blocks) into the position-based dynamics framework.

number of iterations (Deul et al., 2018).

The generalized coordinates of a rod segment $i$ at time $t$ is given by $\mathbf{p}_{i,t} \in \mathbb{R}^3 \times \mathbb{H}$, which includes its position $\mathbf{x}_{i,t} \in \mathbb{R}^3$ given in Cartesian coordinates and its orientation described by a quaternion $\mathbf{q}_{i,t} \in \mathbb{H}$. Correspondingly, $\boldsymbol{v}_{i,t} \in \mathbb{R}^6$ refers to the generalized velocity of the rod segment, which includes velocity and angular velocity. The system is continuously updated during the simulation by applying corrections $\Delta\mathbf{p}_i = (\Delta\mathbf{x}_i, \Delta\boldsymbol{\phi}_i)^\mathsf{T} \in \mathbb{R}^6$ with position shifts $\Delta\mathbf{x}_i \in \mathbb{R}^3$ and orientation shifts $\Delta\boldsymbol{\phi}_i \in \mathbb{R}^3$ representing the integration of the angular velocity.[1]

A single time integration step is presented in Algorithm 1. In the beginning (lines 1 to 4), generalized velocity and generalized coordinates are updated by employing a symplectic Euler integration step. In this regard, $\mathbf{a}_{\text{ext}}$ denotes the generalized acceleration due to the external net force, e.g., given by gravity. XPBD (Macklin et al., 2016) employs the Lagrange multiplier $\boldsymbol{\lambda}$ which is initialized as zero (line 5) along with the integrated generalized coordinates $\mathbf{p}^*$. Collision detection results are stored in $\mathbf{Coll}_{\text{r-r}}$ and $\mathbf{Coll}_{\text{r-p}}$ (line 6), where $\mathbf{Coll}_{\text{r-r}}$ includes all the pairs of two rod segments that potentially collide with each other and $\mathbf{Coll}_{\text{r-p}}$ includes information of all rod segments that potentially collide with another object such as a plane. Within several solver iterations, we alternate between rod constraint projection and the collision constraint projection (lines 7 to 15). The rod constraints include shear-stretch and bend-twist constraints representing the corresponding elastic energy. The Lagrange multiplier represents the interaction between rod segments. Figure 2 illustrates the discretization for a single rod into several interacting segments. The correction values $\Delta\mathbf{p}$ and $\Delta\boldsymbol{\lambda}$ in line 9 are computed by constraint projection (Deul et al., 2018; Kugelstadt & Schoemer, 2016). The generalized coordinates and Lagrange multipliers are updated for each rod (lines 8 to 12), and rod-rod and rod-plane collisions are addressed to update the generalized coordinates. For details about the collision projection procedure, we refer to Macklin et al. (2014). For the non-collision case, the steps within line 6 and 13 are not needed.

The most expensive part of Algorithm 1 involves the computation of the corrections of generalized coordinates and Lagrange multipliers (line 9). This projection step requires the solution of a linear system which is a linearization of a non-linear one so that the matrix depends on the system's state

---

[1]Please note, that $\Delta\mathbf{q}_i = \mathbf{G}(\boldsymbol{q})\Delta\boldsymbol{\phi}_i \in \mathbb{R}^4$, in which the matrix $\mathbf{G}(\boldsymbol{q}) \in \mathbb{R}^{4\times 3}$ describes the relationship between quaternion velocity and angular velocity (Bender et al., 2014a).

---
**Algorithm 1** Numerical integration procedure updating $\mathbf{p}_{i,t} \mapsto \mathbf{p}_{i,t+\Delta t}$ and $\boldsymbol{v}_{i,t} \mapsto \boldsymbol{v}_{i,t+\Delta t}$.
---
1: **for all** rod segments **do**
2:      $\boldsymbol{v}_i^* \leftarrow \boldsymbol{v}_{i,t} + \Delta t\, \mathbf{a}_{\text{ext}}$
3:      $\mathbf{p}_i^* \leftarrow \mathbf{p}_{i,t} + \Delta t\, \mathbf{H}(\mathbf{q}_{i,t})\, \boldsymbol{v}_i^*$ with $\mathbf{H}(\mathbf{q}_{i,t}) := [\mathbf{1}_{3\times3}, \mathbf{0}_{3\times3}; \mathbf{0}_{4\times3}, \mathbf{G}(\mathbf{q}_{i,t})]$
4: **end for**
5: $\boldsymbol{\lambda}^0 \leftarrow \mathbf{0}, \mathbf{p}^0 \leftarrow \mathbf{p}^*$
6: $(\mathbf{Coll}_{\text{r-r}}, \mathbf{Coll}_{\text{r-p}}) \leftarrow \mathsf{generateCollisionConstraints}(\mathbf{p}, \mathbf{p}^*)$
7: **for** $j \leftarrow 0$ to number of required solver iterations **do**
8:      **for all** rods **do**
9:          $(\Delta\mathbf{p}, \Delta\boldsymbol{\lambda}) \leftarrow \mathsf{rodConstraintProjection}(\mathbf{p}^j, \boldsymbol{\lambda}^j)$
10:          $\boldsymbol{\lambda}^{j+1} \leftarrow \boldsymbol{\lambda}^j + \Delta\boldsymbol{\lambda}$
11:          $\mathbf{p}^{j+1} \leftarrow \mathbf{p}^j + \Delta\mathbf{p}$
12:      **end for**
13:      $\mathbf{p}^{j+1} \leftarrow \mathsf{updateCollisionConstraintProjection}(\mathbf{p}^{j+1}, \mathbf{Coll}_{\text{r-r}}, \mathbf{Coll}_{\text{r-p}})$
14:      $j \leftarrow j + 1$
15: **end for**
16: **for all** rod segments **do**
17:      $\mathbf{p}_{i,t+\Delta t} \leftarrow \mathbf{p}_i^j$
18:      $\boldsymbol{v}_{i,t+\Delta t} \leftarrow \mathbf{H}^\mathsf{T}(\mathbf{q}_{i,t})(\mathbf{p}_{i,t+\Delta t} - \mathbf{p}_{i,t})/\Delta t$
19: **end for**
---

making it impossible to precompute its inverse. Instead, a new system at every point in time is solved iteratively using the conjugated gradient (CG) solver. Such iterative solvers are widely used in the context of physical simulations and regularly described as the de facto standard (Barrett et al., 1994; Saad, 2003) since they often show superior performance and usually scale well allowing for exploiting parallel hardware. However, we would like to point out that also highly efficient direct solvers can be found in the literature (Deul et al., 2018).

Instead of fully replacing the projection step in an end-to-end learning manner, we follow the strategy of accelerating it by first computing a guess

$$(\Delta\mathbf{p}^0, \Delta\boldsymbol{\lambda}^0) \leftarrow \mathsf{correctionGuess}(\mathbf{p}^j)\,, \tag{1}$$

for the iterative procedure (line 9)

$$(\Delta\mathbf{p}, \Delta\boldsymbol{\lambda}) \leftarrow \mathsf{rodConstraintProjection}(\mathbf{p}^j, \boldsymbol{\lambda}^j, \Delta\mathbf{p}^0, \Delta\boldsymbol{\lambda}^0)\,. \tag{2}$$

A neural network is employed to compute the initial guess in Eq. (1) for the constraint projection. The motivation for this approach is to reduce the number of iterations required for the convergence of the CG solver which solves the linear system in Eq. (2) compared to the canonical initialization with zeros. We obtain our final framework by replacing line 9 in Algorithm 1 with Eq. (1) and Eq. (2) as illustrated in Figure 3, which is inherently as accurate as the traditional PBD method. We name the data-driven part *COPINGNet* ("COnstraint Projection INitial Guess Network") which learns to compute the correction guess.

## 3.1 Graph Encoding

COPINGNet is a graph network based architecture which we apply in order to compute initial guesses for $\Delta\mathbf{p}$ and $\Delta\boldsymbol{\lambda}$. In this regard, we need to incorporate the rods' state into a graph description (Battaglia et al., 2018). A graph $G = (V, E, U)$ usually consists of nodes (or vertices) $V$, edges $E$ as well as global features $U$. However, in our framework, global features $U$ are not used. For example, gravity could be a potentially meaningful global feature, but it also can be easily included as an external acceleration. Hence, $U = \emptyset$ and the graph can just be represented as $G = G(V, E)$. In our case, the rods' segments within the scene are represented by the graph's nodes while the interactions between the rods' segments are represented by the edges.
COPINGNet provides a graph-to-graph mapping: $\mathbb{G}^{\text{in}} \rightarrow \mathbb{G}^{\text{out}}$, from an input graph $G_{\text{in}} \in \mathbb{G}^{\text{in}}$ to an output graph $G_{\text{out}} \in \mathbb{G}^{\text{out}}$. Nodes and edges of both graphs are equipped with specific features. In the case of the input graph, the node features describe the state of the rods' segments, i.e.

$$\mathbf{v}_{\text{in},i} = (\mathbf{x}_i, \mathbf{q}_i, r_i, \rho_i, \ell_i, \alpha_i, f_{0_i}, f_{1_i}, f_{2_i})^\mathsf{T} \in \mathbb{V}^{\text{in}} \subseteq \mathbb{R}^{14}\,,$$

in which the positions are denoted with $\mathbf{x}_i \in \mathbb{R}^3$, the orientations with $\mathbf{q}_i \in \mathbb{H}$, the radii with $r_i \in \mathbb{R}^{>0}$, and the densities with $\rho_i \in \mathbb{R}^{>0}$, and the segment lengths with $\ell_i \in \mathbb{R}^{>0}$. Moreover, a segment position indicator $\alpha_i \in [0,1] \subset \mathbb{R}$ is included corresponding to a parameterization by arc length.[2] Binary segment flag $f_{0_i} \in \{0,1\}$, "left" end flag $f_{1_i} \in \{0,1\}$ and "right" end flag $f_{2_i} \in \{0,1\}$ are set to zero if the specific segment respectively the left or right segment of the rod is fixed and to one otherwise. The nodes of $G_\mathsf{in}$ are given as the set of $V_\mathsf{in} = \cup_{i=1}^{n}\{\mathbf{v}_{\mathsf{in},i}\}$, in which $n = |V_\mathsf{in}|$ denotes the number of segments in the scene. The nodes of $G_\mathsf{out}$ contain the correction values of the generalized coordinates, i.e.

$$\mathbf{v}_{\mathsf{out},i} = \Delta\mathbf{p}_i \in \mathbb{V}^\mathsf{out} \subseteq \mathbb{R}^6 \text{ and } V_\mathsf{out} = \cup_{i=1}^{n}\{\mathbf{v}_{\mathsf{out},i}\}.$$

While rod segments are represented as node features, we represent constraints between rod segments as edge features:

$$\mathbf{e}_{\mathsf{in},i} = (\boldsymbol{\omega}_i, Y_i, T_i)^\mathsf{T} \in \mathbb{E}^\mathsf{in} \subseteq \mathbb{R}^5,$$

in which the (rest) Darboux vector $\boldsymbol{\omega} \in \mathbb{R}^3$ describes the static angle of two rod segments, and Young's modulus $Y \in \mathbb{R}^{>0}$ and torsion modulus $T \in \mathbb{R}^{>0}$ are corresponding to extension, bending, and twist constraint parameters. The set of edges of the input graph is then given by $E_\mathsf{in} = \cup_{i=1}^{m}\{\mathbf{e}_{\mathsf{in},i}\}$, in which $m = |E_\mathsf{in}|$ denotes the number of interactions between different segments. The correction of the Lagrange multiplier $\Delta\boldsymbol{\lambda}_i \in \mathbb{R}^6$ is stored in the output edges:

$$\mathbf{e}_{\mathsf{out},i} = \Delta\boldsymbol{\lambda}_i \in \mathbb{E}^\mathsf{out} \subseteq \mathbb{R}^6.$$

The set of output edges is then given by $E_\mathsf{out} = \cup_{i=1}^{m}\{\mathbf{e}_{\mathsf{out},i}\}$.
The connectivity information $C$ of each graph is stored in two vectors $\mathbf{c}_\mathsf{sd}$ and $\mathbf{c}_\mathsf{rv}$ containing the sender node index and the receiver node index of each corresponding edge. This concludes the specification of the input graph $G_\mathsf{in} = G(V_\mathsf{in}, E_\mathsf{in})$ and the output graph $G_\mathsf{out} = G(V_\mathsf{out}, E_\mathsf{out})$ with connectivity information $C = (\mathbf{c}_\mathsf{sd}, \mathbf{c}_\mathsf{rv})$.

## 3.2 Network Structure

In the following, we describe the structure of COPINGNet after we formalized its input and output in the previous section. As illustrated in Figure 3, the network consists of an encoder network, multiple stacks of GN-blocks, and a decoder network. The graph network from Battaglia et al. (2018) is used as a basic element and denoted as a GN-block. Residual connection between blocks could improve performance of neural networks in both CNN (He et al., 2015), and graph neural network [Li et al. (2019)]. As in related work (Sanchez-Gonzalez et al., 2020), we employ the residual connection between the GN-blocks, but our encoder/decoder network directly deals with the graph. The encoder network performs a mapping: $\mathbb{G}^\mathsf{in} \rightarrow \mathbb{G}^\mathsf{latent}$ and is implemented using two multi-layer perceptrons (MLPs), $\mathsf{MLP}_\mathsf{edge} : \mathbb{E}^\mathsf{in} \rightarrow \mathbb{E}^\mathsf{latent} \subseteq \mathbb{R}^l$ and $\mathsf{MLP}_\mathsf{node} : \mathbb{V}^\mathsf{in} \rightarrow \mathbb{V}^\mathsf{latent} \subseteq \mathbb{R}^l$, in which $l$ denotes the latent size. They work separately and thus $E_\mathsf{en} = \mathsf{MLP}_\mathsf{edge}(E_\mathsf{in})$ and $V_\mathsf{en} = \mathsf{MLP}_\mathsf{node}(V_\mathsf{in})$. Edge features $E_\mathsf{in}$ from the input are constant for a rod during the simulation and this results in constant encoded edge features $E_\mathsf{en}$, which could be recorded after the first run and used afterwards during inference. The edge feature $\mathbf{e}_{\mathsf{in},i}$ contains the material parameters which could vary by different orders of magnitude. Hence, we normalize Young's modulus and torsion modulus before feeding them into the network. After encoding, the graph $G_\mathsf{en}(V_\mathsf{en}, E_\mathsf{en}) \in \mathbb{G}^\mathsf{latent}$ is passed to several GN-blocks with residual connections. Each GN-block also contains two MLPs. However, they use message passing taking advantage of neighbourhood nodes/edges information (Battaglia et al., 2018). A number of $M$ GN-blocks enable the use of neighbourhood information with distances smaller or equal to $M$. The graph network performs a mapping within the latent space: $\mathbb{G}^\mathsf{latent} \rightarrow \mathbb{G}^\mathsf{latent}$, and after $M$ GN-blocks, we obtain $G'_\mathsf{en}(V'_\mathsf{en}, E'_\mathsf{en}) \in \mathbb{G}^\mathsf{latent}$. The decoder network performs a mapping: $\mathbb{G}^\mathsf{latent} \rightarrow \mathbb{G}^\mathsf{out}$, which has a similar structure as the encoder network. Two MLPs $\mathsf{MLP}_\mathsf{edge} : \mathbb{E}^\mathsf{latent} \rightarrow \mathbb{E}^\mathsf{out}$ and $\mathsf{MLP}_\mathsf{node} : \mathbb{V}^\mathsf{latent} \rightarrow \mathbb{V}^\mathsf{out}$ compute $E_\mathsf{out} = \mathsf{MLP}_\mathsf{edge}(E'_\mathsf{en})$ and $V_\mathsf{out} = \mathsf{MLP}_\mathsf{node}(V'_\mathsf{en})$. A $\tanh$-function at the end of $\mathsf{MLP}_\mathsf{edge}$ and $\mathsf{MLP}_\mathsf{node}$ is used restricting the output to the interval $[-1,1] \subset \mathbb{R}$. COPINGNet learns the relative correction values. The generated dataset is normalized, and the maximum correction value of the generalized coordinates and the Lagrange multipliers will be recorded as norms. The final correction value is achieved by multiplying the relative correction values and the norms. This normalization process damps the

---

[2]For a single rod in the scene which consists of $N$ segments of equal lengths, for the $i$-th segment, we obtain $\alpha_i = (i-1)/(N-1)$ for $i \in \{1,2,\dots,n\}$.

| Train/Val | #Steps | #Nodes $N$ | Young's Modulus $Y$ | Initial Angle $\phi_0$ | Rod Length $\ell$ |
|---|---|---|---|---|---|
| 256/100 | 50 | $\mathcal{U}_\mathrm{d}(10,55)$ | $10^a$ Pa, $a \sim \mathcal{U}(4.0,6.0)$ | $\mathcal{U}(0°,45.0°)$ | $\mathcal{U}(1.0\text{ m}, 5.0\text{ m})$ |
| **Train/Val** | **#Steps** | **#Nodes $N$** | **Torsion Modulus $T$** | **Helix Radius / Height** | **Winding Number** |
| 256/100 | 50 | $\mathcal{U}_\mathrm{d}(45,105)$ | $10^a$ Pa, $a \sim \mathcal{U}(4.0,6.0)$ | $\mathcal{U}(0.4\text{ m}, 0.6\text{ m}) / \mathcal{U}(0.4\text{ m}, 0.6\text{ m})$ | $\mathcal{U}(2.0,3.0)$ |

Table 1: Specification of training and validation datasets for the two scenarios of an initially straight bending rod (top) and an elastic helix (bottom). The datasets are comprised of a number of data points (left) each describing the rod's dynamics within $t \in [0\text{ s}, 50\Delta t]$ discretized with a time step size of $\Delta t = 0.02$ s. The number of nodes $N$ is sampled from a discrete uniform distribution $\mathcal{U}_\mathrm{d}$ while the remaining parameters are sampled from a continuous uniform distribution $\mathcal{U}$. We trained our network for 5 hours for the bending rod scenario and 6 hours for the helix case.

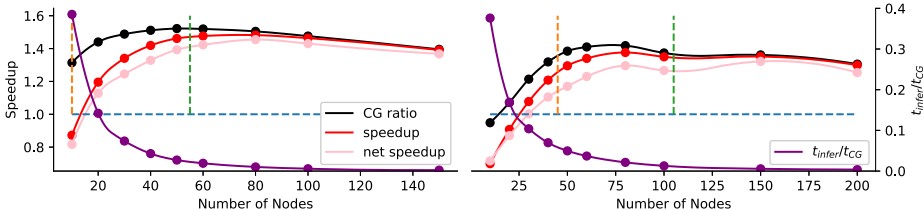

Figure 4: Illustration of the ratio of COPINGNet's inference time $t_\mathrm{infer}$ and the vanilla CG solver's run time $t_\mathrm{CG}$ (purple curves; right vertical axis) for the initially straight bending rod (left) and the elastic helix (right) simulations. The black curves show the CG iteration number ratio while the red curves show the total speedup of the constraint projection when taking into account COPINGNet's inference time (left vertical axis). The pink curves show the total speedup of the entire simulations. The orange and green dashed line indicate the lower and upper boundaries of the total number of nodes used in the training data. We can observe a speedup even for rods and helices with greater node number than the ones used in the the training dataset. The result is averaged from 50 simulations each running 100 steps.

noise caused by the network and leads to a more stable performance. For simplicity, all the MLPs in different blocks have the same number of layers, and the same width as latent size $l$ within the latent layers. The input and output sizes of each MLP are consistent with the corresponding node/edge feature dimensions. The loss $L$ is computed from both parts, nodes and edges,

$$L := \mathsf{MSE}(V_\mathrm{out}, \tilde{V}_\mathrm{out}) + \mathsf{MSE}(E_\mathrm{out}, \tilde{E}_\mathrm{out}),$$

in which $(V_\mathrm{out}, E_\mathrm{out})$ is the output graph's ground truth in contrast to COPINGNet's prediction $(\tilde{V}_\mathrm{out}, \tilde{E}_\mathrm{out})$. The mean squared error between $\chi$ and $\tilde{\chi}$ is denoted with $\mathsf{MSE}(\chi, \tilde{\chi})$.

For evaluation purposes, we also incorporated the $k$-nearest neighbor ($k$-NN) algorithm as described in the supplementary material setting $k = 3$.

# 4   Evaluation

We generate training and validation datasets based on two scenarios: an initially straight bending rod and an elastic helix each fixed at one end oscillating under the influence of gravity. The specification of these datasets is provided in Table 1. The PBD code is written in C++, while the COPINGNet is implemented in PyTorch. The training is performed on an NVIDIA® Tesla® V100 GPU. The training time varies from 4 to 30 hours for different architecture parameters. A constant learning rate of $\eta = 0.001$ was used and a mean square loss function was employed. Our approach generalizes across different initial conditions, geometries, discretizations, and material parameters. In the supplementary material we show that it is possible to robustly generate various dynamical results (Figure 9). For a discussion on the network architecture please see Figure 11 (supplementary material).

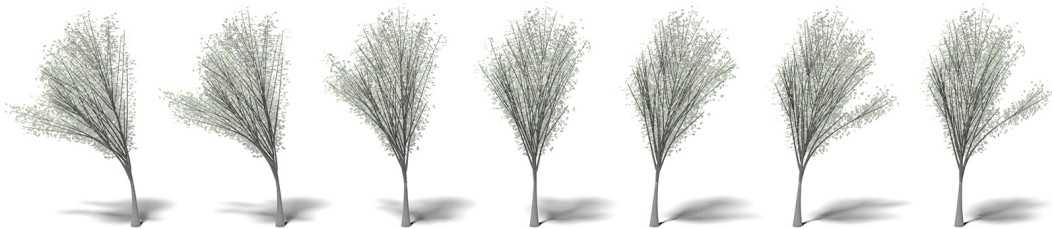

Figure 5: Realistic biomechanical simulation of a 3D tree model composed of over $1k$ nodes swaying in a wind field. Our GN approach performs correctly even under a large number of rod segments while increasing performance of the original PBD method.

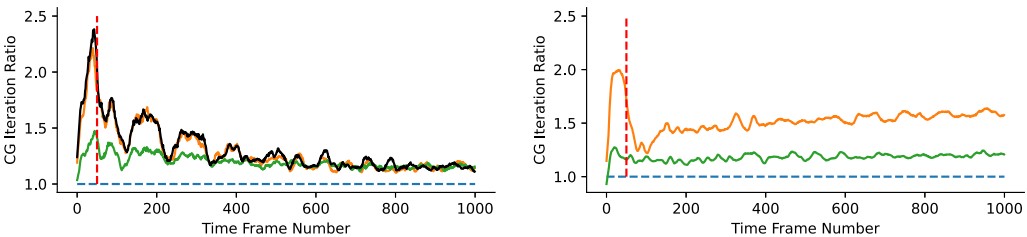

Figure 6: The dashed horizontal lines show the benchmark CG solver's constant performance for an initially straight bending rod simulation (left) and an elastic helix simulation (right). We are comparing our COPINGNet-assisted PBD approach (orange) to a simplified version in which a $k$-NN (green) method is used to predict the initial guess. Moreover (left), we added a performance measurement in which we restricted the solution to be in a two-dimensional plane (black). The results are averages over 20 simulations and smoothed with a window of 10 frames. The dashed vertical lines mark the number of 50 frames contained in the training data set. For the helix simulation we observe a speed up of approximately 50% even beyond the training data (50 time frames). For the bending rod simulation we observe a speedup that further decreases with increasing time frame number.

**Discretization.** Our approach addresses the acceleration of the most expensive part within PBD by providing an accurate initial guess of the constraint projection. We measure the system's complexity by the number of nodes in a rod. Figure 4 shows the ratio of COPINGNet's inference run time (black and red curves) compared to the run time of the vanilla CG solver (purple curves) for different numbers of nodes. The increasing black and red curves indicate that the speedup of COPINGNet is more significant with greater number of nodes. With only a few nodes the CG solver performs better due to the inference overhead. Once the number of nodes is increasing, a significant speedup can be obtained of up to 50% for the constraint projection. Surprisingly, our approach also performs well when going far beyond the sampling range (orange to green dashed lines). Since the constraint projection is the most time-consuming part of the entire simulation, the speedup converges to that of the whole simulation (pink curves) with increasing number of nodes as shown in Figure 4.

**Temporal Evolution.** In addition to the complexity analysis, we also analyze the required number of CG iterations for the vanilla constraint projection compared to the one accelerated using COPINGNet over time as shown in Figure 6. We obtain a significant total speedup compared to the CG solver (dashed blue line). As stated in Table 1, our training data contains dynamical simulations of 50 time steps. In this range we observe the highest speedup. As was the case for the complexity analysis, we again obtain a significant speedup for simulations beyond 50 time steps. Compared to a $k$-NN (green) method used to predict the initial guess, our COPINGNet-assisted PBD approach shows the best performance in both scenarios.

**Long-term Stability.** In case the constraint projection is completely replaced by COPINGNet (end-to-end approach), stability of the PBD method decreases as error accumulation takes place. This is

illustrated in Figure 7 showing the temporal evolution of the relative change of the total rod length. An initially straight rod bending under the influence of gravity is simulated using the parameters $\phi_0 = 0°$, $N = 30$, $\ell = 4.0$ m, and varying Young's modulus $Y$. In this scenario the rod length is expected to stay constant during the mechanical simulation. In case COPINGNet is used in an end-to-end manner (colored lines), where the whole constraint projection step is replaced, we observe that the rod length changes incorrectly. In fact, the divergence of using COPINGNet to replace the constraint projection step increases exponentially beyond the range of the training data (50 steps). To the contrary, when COPINGNet is used to only estimate the initial guess of the constraint projection (black lines), no rod deformation takes place even beyond the training data range.

**Collisions.** Figure 1 illustrates a collision of an elastic helix with the ground plane and a collision between two rods. Collision detection is efficiently implemented using the hierarchical spatial hashing scheme according to Eitz & Gu (2007). Rod-rod collisions are then treated with line-to-line distance constraints, and collisions with the ground plane using half-space constraints. Moreover, frictional effects are implemented according to Macklin et al. (2014). This approach allows us to handle discontinuous events such as collisions between individual rods and other objects. Moreover, for this experiment we use the neural networks trained on the rod and helix simulations. Employing COPINGNet trained on the helix data to simulate the collision of the ground plane (HR = HH = 0.5 m, HW = 2.5, $G = 1.0 \cdot 10^6$ Pa, $N = 50$), we measure a total speedup of approx. 10%. In the case of two colliding rods ($\phi_0 = 0°$, $N \in \{20, 30\}$, $\ell \in \{4.0\text{ m}, 4.5\text{ m}\}$, $Y = 1.0 \cdot 10^6$ Pa), we obtain a speedup of approx. 6%.

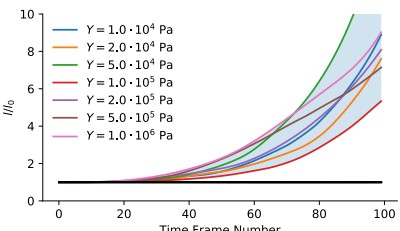

Figure 7: Illustration of the relative change of the total rod length ($l/l_0$) for different values of Young's moduli ($Y$). Colored lines show different results for COPINGNet replacing the constraint projection. The thick black line represent the result for COPINGNet replacing only the initial guess. This indicates that the end-to-end approach does not generalize past the 50 time steps used in preparing the training data.

**Complex Scenarios.** Our method is also capable to deal with complex scenarios such as 3D tree models swaying in wind fields as shown in Figure 5. We represent trees using the extended Cosserat rod theory introduced in Pirk et al. (2017). This method allows us to simulate realistic biomechanics of thousands of rod segments at interactive rates. We generated 100 different tree topologies using 70 individual rods (average node number: 1056) and simulate the swaying motions of these tree models with vanilla PBD to generate the training dataset. For the evaluation, 30 different tree topologies have been generated for each of the following four experiments using 10 rods (204 nodes), 20 rods (355 nodes), 40 rods (654 nodes), and 70 rods (1061 nodes). We were able to significantly improve the runtime performance of the method by 17.0% (10 rods), 15.8% (20 rods), 13.0% (40 rods), and 11.1% (70 rods). This takes into account the inference time introduced by the neural network.

**Generalization.** Figure 7 and Figure 10 (supplementary material) indicate that using COPINGNet in an end-to-end manner does not generalize beyond the training data. Specifically, the end-to-end setup diverges in terms of rod geometry and segment position from the correct solution. This effect increases significantly beyond the training data range. Although, we only use COPINGNet as a benchmark for this evaluation, other GNs are expected to perform similarly. Common workarounds to increase the stability of dynamical systems with neural networks are temporal skip-connections, recurrent training, and data augmentation. However, these approaches focus on runtime speed and memory performance rather than stability (Holden et al., 2019). Interestingly, replacing just the initial guess with a GN does not deform the rod or introduces discontinuities at any observed stage of the bending simulation. This means that employing GNs can result in a performance increase without a loss of stability. Furthermore, a GN that only provides initial guesses seems to also generalize to other scenarios. We observed performance improvements for the collision and complex tree case, although the network was never trained on these different rod discretizations and topologies. This indicates that our method is capable to generalize within a specific physical scenario and to a lesser extent to other scenarios.

**Limitations.** Apart from being useful for accelerating the simulation of mechanical systems, e.g., using iterative methods such as in position-based dynamics, finite elements analysis or in the con-

text of linear complementarity problems, we see a broader impact of our work in contributing to the further development of synergy effects of machine learning (e.g. graph learning) and physical simulations. However, in this contribution we do not systematically explore this potential as we have restricted ourselves to rod simulation using position-based dynamics. In future work, we plan to study our approach in the context of other iterative procedures in physics-based modeling and simulation. With respect to the simulation of dynamic rods, the speedup achieved from our framework is reduced in complex case when collisions are involved. Collision handling is another mainstream topic, e.g., within the computer graphics community. We plan to adapt finding from previous work within the graphics community to enhance the degree of sophistication of our approach when it comes to complex collision handling.

## 5 Conclusion

We discovered that applying GNs for replacing the initial guess has fundamental advantages over end-to-end approaches. First, our network-enabled solver ensures long-term stability inherited from traditional solvers for physical systems, while improving runtime performance. Second, our approach is able to generalize across different initial conditions, rod discretizations, and material parameters, and it handles discontinuous effects such as collisions. While end-to-end approaches offer more significant speedups, our method is superior in cases where stability is an essential requirement.

Our approach to accelerate iterative solvers with GNs opens multiple avenues for future work. For one, it would be interesting to explore mechanical systems describing general deformable (e.g. textiles) or volumetric objects, which have been simulated with PBD. Second, our approach can be applied to other iterative methods, such as in finite elements analysis or in the context of linear complementarity problems (LCP). This would allow us to accelerate physical simulations, when iterative solvers are applied, without compromising stability. From a more general perspective, other learning techniques can also be explored in the context of rod dynamics such as reinforcement or imitation learning (Li et al., 2018a; Müller et al., 2017).

## Acknowledgements

This work was supported and funded by KAUST through the baseline funding of the Computational Sciences Group and a Center Partnership Fund of the Visual Computing Center. The valuable comments of the anonymous reviewers that improved the manuscript are gratefully acknowledged.

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
