# Supplementary Material

## Dynamic Results

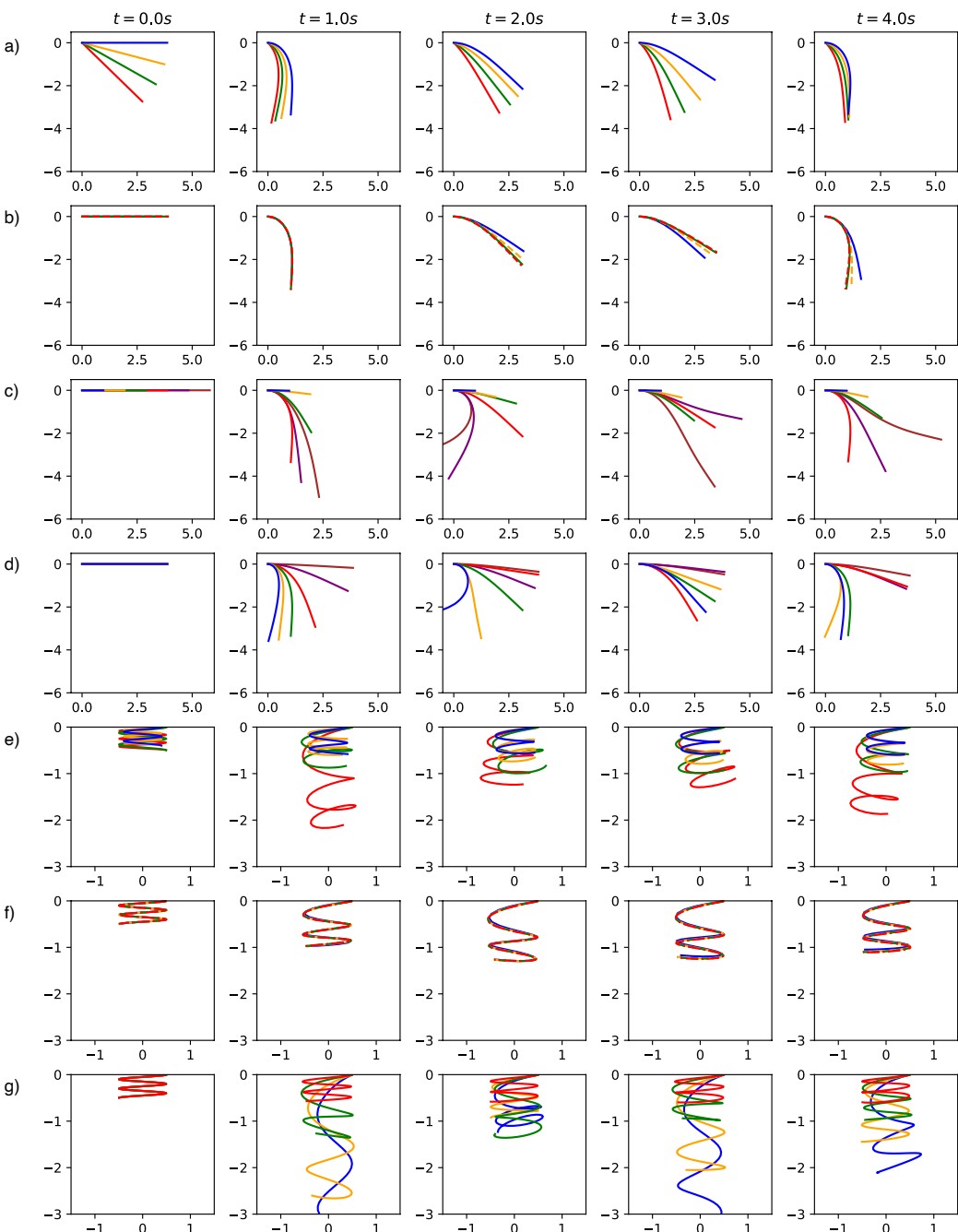

Figure 8: Illustration of the temporal evolution of scenarios (a) to (d) for the initially straight bending rod, and (e) to (f) for the elastic helix. The different cases represent various material property configurations.

In Figure 8 we show the temporal evolution of different scenarios (a) to (d) for the initially straight bending rod, and (e) to (f) for the elastic helix. The default parameters of the initially straight bending rod are $\phi_0 = 0°$, $N = 30$, $\ell = 4.0$ m, and $Y = 1.0 \cdot 10^5$ Pa. In (a), we modify $\phi_0 \in \{0.0°, 15.0°, 30.0°, 45.0°\}$. In (b), we modify $N \in \{10, 20, 40, 60\}$. In (c), we modify $\ell \in \{1.0\,\text{m}, 2.0\,\text{m}, 3.0\,\text{m}, 4.0\,\text{m}, 5.0\,\text{m}, 6.0\,\text{m}\}$. In (d), we modify $Y \in \{2.0 \cdot$

$10^4\,\mathrm{Pa}, 5.0 \cdot 10^4\,\mathrm{Pa}, 1.0 \cdot 10^5\,\mathrm{Pa}, 2.0 \cdot 10^5\,\mathrm{Pa}, 5.0 \cdot 10^5\,\mathrm{Pa}, 1.0 \cdot 10^6\,\mathrm{Pa}\}$. The default parameters of the elastic helix are $\mathsf{HR} = 0.5$ m (helix radius), $\mathsf{HH} = 0.5$ m (helix height), $\mathsf{HW} = 2.5$ (winding number), $T = 1.0 \cdot 10^5$ Pa, and $N = 60$. In (e), we modify $(\mathsf{HR}, \mathsf{HH}, \mathsf{HW}) \in \{(0.4\,\mathrm{m}, 0.4\,\mathrm{m}, 2.0\,\mathrm{m}), (0.4\,\mathrm{m}, 0.4\,\mathrm{m}, 3.0\,\mathrm{m}), (0.5\,\mathrm{m}, 0.5\,\mathrm{m}, 2.0\,\mathrm{m}), (0.5\,\mathrm{m}, 0.5\,\mathrm{m}, 3.0\,\mathrm{m})\}$. In (f), we modify $N \in \{30, 60, 80, 100\}$. In (g), we modify $G \in \{2.0 \cdot 10^4\,\mathrm{Pa}, 5.0 \cdot 10^4\,\mathrm{Pa}, 1.0 \cdot 10^5\,\mathrm{Pa}, 1.0 \cdot 10^6\,\mathrm{Pa}\}$. For each experiment, the rod colors indicate the corresponding parameters in the following (ascending) order: blue, orange, green, red, purple, brown.

These results are obtained by employing three GN blocks, two MLP layers, and a MLP width of 32. Parameters have been used which are are not covered within the training dataset.

Moreover, to further demonstrate general applicability, Figure 9 shows an additional simulation of a tightening knot for which COPINGNet-based PBD shows a total speedup of $10.4\%$ compared to vanilla PBD. COPINGNet is trained using a dataset containing 20 simulations of the same knot scenario with different discretizations over 100 frames. It can clearly be observed that COPINGNet generalizes beyond the training data.

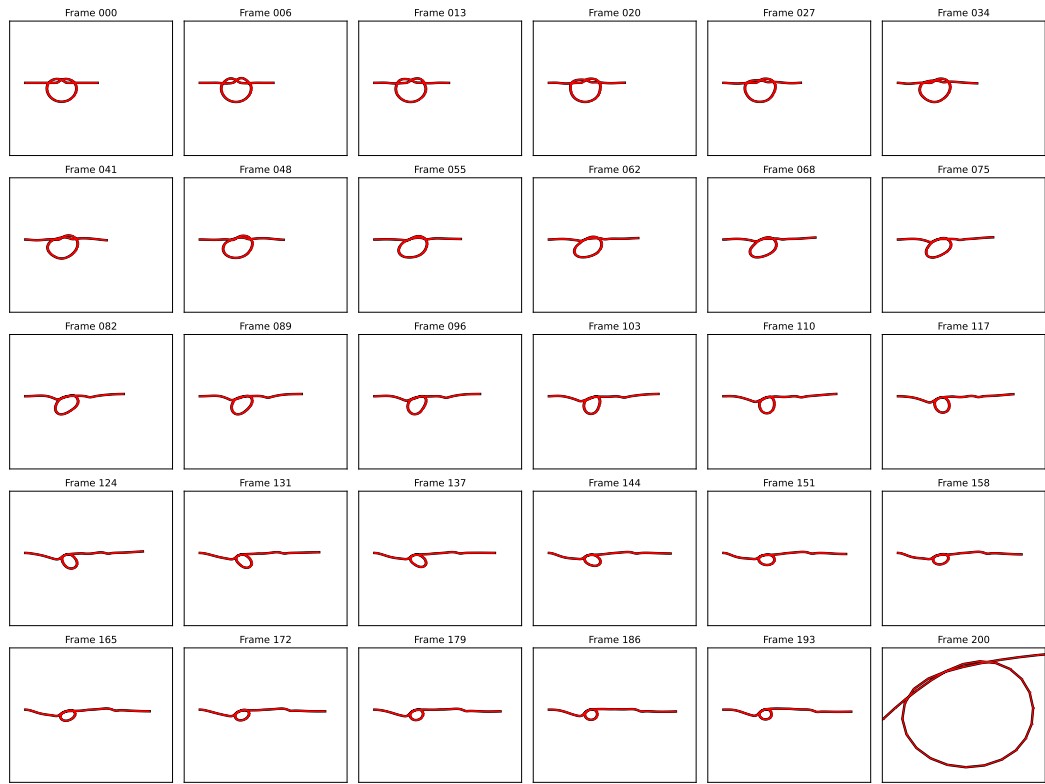

Figure 9: Illustration of the temporal evolution of a knot scenario. The rod is fixed at both ends and the knot is pulled tight. The predicted results by end-to-end COPINGNet-based learning are shown in red while the results computed with COPINGNet-assisted PBD are shown in black. The visualization of the last frame contains a close-up to demonstrate that the knot structure is still preserved. The rod parameters are $N = 110$, $R = 0.01$ m, $\ell = 11.0$ m, and $Y = T = 1.0 \cdot 10^4$ Pa.

**Comparisons**

Figure 10 illustrates the temporal evolution of a bending rod and elastic helix scenarios using different approaches. For the bending rod case, the parameters $\phi_0 = 0°$, $N = 30$, $\ell = 4.0$ m, and $Y = 1.0 \cdot 10^5$ Pa are used. In the helix case, $\mathsf{HR} = 0.5$ m (helix radius), $\mathsf{HH} = 0.5$ m (helix height), $\mathsf{HW} = 2.5$ (winding number), $T = 1.0 \cdot 10^5$ Pa, and $N = 60$ were applied. The temporal evolution of the positions' normalized root mean square error (NRMSE) is shown in Figure 11.

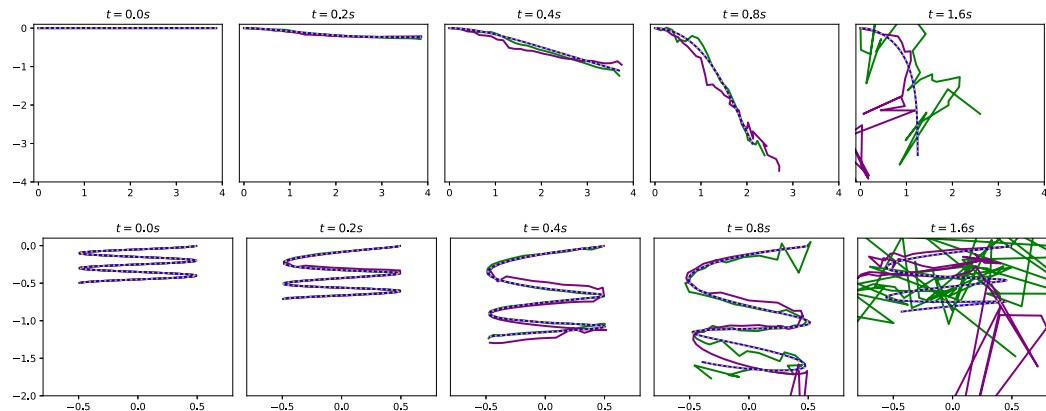

Figure 10: Illustration of the temporal evolution of bending rod (upper row) and elastic helix (lower row) scenarios simulated using our COPINGNet-assisted PBD approach (blue curves), vanilla PBD (orange dotted curves), $k$-NN-based end-to-end learning (purple curves) and GN-based end-to-end learning (green curves). While the COPINGNet-assisted approach and the vanilla PBD solver allow us to simulate both cases in a stable manner, both end-to-end learning approaches diverge with increasing time due to error accumulation.

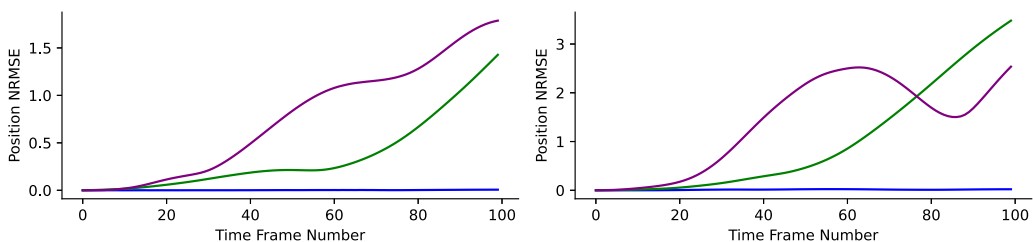

Figure 11: Illustration of the positions' normalized root mean square error (NRMSE) compared to vanilla PBD using our COPINGNet-assisted PBD approach (blue curves), $k$-NN-based end-to-end learning (purple curves), and GN-based end-to-end learning (green curves) for the bending rod (left) and the elastic helix (right) scenarios. While the COPINGNet-assisted PBD approach converges to an almost correct solution, both end-to-end learning approaches show large error rates with increasing time frame number.

**Architecture**

The evaluation of COPINGNet's architecture, introduced in Section 3.2, is presented in Figure 12. The performance is studied for different numbers of GN-blocks and MLP layers, and different MLP widths (latent sizes). The measurements demonstrate that a larger number of GN-blocks usually increases the performance while the performance improvement is not longer significant for more than four GN-blocks. This is plausible and can be considered analogously to the size of the stencil of a numerical integrator: a larger number of GN-blocks means that a specific node can take advantage of information gained from neighbourhoods further away. Correspondingly, a larger stencil can do so as well. However, increasing the size of the stencil usually does not result in more accurate results once the critical size is reached. This can be observed here as well. In contrast, it can be clearly observed that deeper MLP networks do not improve the performance. This is consistent with other research on graph networks (Sanchez-Gonzalez et al., 2020). However, increasing the MLP width can increase the network's ability to generalize. It is shown that the highest performance is achieved with the largest MLP width, especially in the case of the helix scenarios. Since increasing the number of GN-blocks and MLP width will lead to longer inference time, we compromise by choosing medium numbers. For further evaluations, we employ three GN-blocks, two MLP layers, and a MLP width of 32.

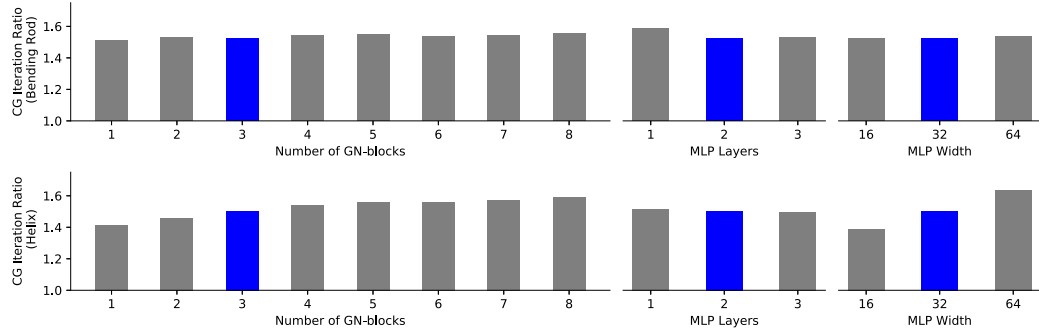

Figure 12: Illustration of the evaluation of COPINGNet's architecture (blue: benchmark architecture). The result is averaged from $50$ test simulations each running for $100$ time steps.

### $k$-nearest Neighbor Algorithm

For evaluation purposes, we also incorporated the $k$-nearest neighbor ($k$-NN) algorithm into our framework to compute an initial guess for the constraint projection step.

Consider a $k$-NN data point $(\mathbf{x_i}, \mathbf{y_i})$. Its components contain node and edge feature information:

$$\mathbf{x}_i = \left(\mathbf{v}_{\mathsf{in},i}, \mathbf{e}_{\mathsf{in},i_k}, \dots\right)^{\mathsf{T}}, \quad \mathbf{y}_i = \left(\mathbf{v}_{\mathsf{out},i}, \mathbf{e}_{\mathsf{out},i_k}, \dots\right)^{\mathsf{T}}, k \in \mathbb{N}_0 \,.$$

Simulation data is collected to form a dataset $\{(\mathbf{x}_i, \mathbf{y}_i)\}$. During inference, the nearest $k$ data points of $\mathbf{x}_i^*$ are picked and linear interpolations are performed to obtain the predicted $\mathbf{y}_i^*$.

For each input node feature $\mathbf{v}_{\mathsf{in},i}^*$, we obtain the corresponding output node feature $\tilde{\mathbf{v}}_{\mathsf{out},i} = \mathbf{v}_{\mathsf{out},i}^*$. The output edge feature is averaged from the information of the corresponding component $\mathbf{y}_j^*$:

$$\tilde{\mathbf{e}}_{\mathsf{out},i} = \frac{1}{M} \sum_j \mathbf{e}_{\mathsf{out},j_k}^* \,,$$

in which $M$ denotes the number of nodes.

The $k$-NN algorithm is applied as an end-to-end method completely replacing constraint projection or to predict an initial guess.

### Inhomogeneous Material

We also conducted experiments in which we simulated inhomogeneous material. In this setup, we consider a rod initially arranged horizontally with a fixed left end. From left to right, the Young's modulus is linearly interpolated from $1.0 \cdot 10^6$ Pa to a minimum value taken from $\{1.0 \cdot 10^4$ Pa, $5.0 \cdot 10^4$ Pa, $1.0 \cdot 10^5$ Pa, $5.0 \cdot 10^5$ Pa$\}$. The model is trained using homogeneous material data as in the previous section. For the four scenarios we have measured a CG iteration speedup compared to the vanilla case of $20.6\%, 21.8\%, 21.3\%$, and $21.7\%$. We have also monitored the relative length change with respect to the end-to-end case. Figure 13 shows the results for COPINGNet replacing the constraint projection.

### Diagonal Preconditioned Conjugate Gradient

We also tested the widely used diagonal preconditioned conjugate gradient (DPCG) method. As expected, it turned out that DPCG is unable to reduce the number of iterations required for convergence as the system matrix of our problem is not diagonally dominant.

### Coarse Grid Approach

Finally, we also evaluated a coarse grid approach to compute the initial guess for simulating with a higher resolution. In our setup, a bending rod with a length of $4.0$ m is discretized using $10$ rod segments (coarse case) and using $40$ rod segments (detailed case). Within each integration step of the detailed case, we apply a linear interpolation considering the results of the coarse case. These

results are used as the initial guess for the detailed case. We run an experiment with different Young's moduli of $1.0 \times 10^4$ Pa, $5.0 \times 10^4$ Pa, $1.0 \times 10^5$ Pa, $5.0 \times 10^5$ Pa, and $1.0 \times 10^5$ Pa. While our COPINGNet method achieves an average net speedup of $30.6\%$, this method achieves an average speedup of $45.5\%$ if the overhead is not taken into account. However, this method requires additional computation time for the simulation with a lower resolution resulting in a net speedup of less than $20\%$.

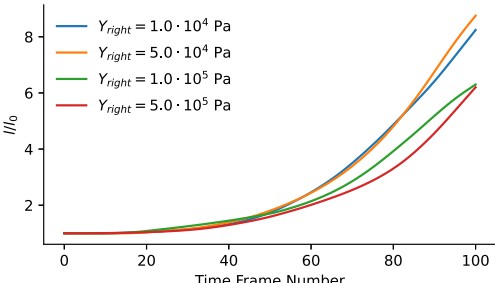

Figure 13: Illustration of the relative change of the total rod length ($l/l_0$) for different values of the right end Young's moduli ($Y$). The curves show different results for COPINGNet replacing the constraint projection part.