# OpenReview forum: "Accurately Solving Rod Dynamics with Graph Learning"
_NeurIPS.cc/2021/Conference — NeurIPS 2021 Poster_

### Official Review · Reviewer_5qbT · 2021-07-12

**Rating:** 6
**Confidence:** 3

**Summary:**

This paper proposes to train a graph network to estimate the initial guess of a conjugate gradient solver in a rod dynamics simulation pipeline. By using a better initial position given by the graph network, the CG solver can converge faster than optimizing from zero vectors. The authors show how to construct such a network and demonstrate the accuracy and efficiency of their hybrid method in various simulation settings.

**Limitations And Societal Impact:**

I don’t think there would be serious negative societal impacts in this work. The limitation of this work I think is its significance and real-world use, which I covered in the last part.

**Main Review:**

Originality:
Yes. This is my first timing reading accelerating the simulation of rod dynamics using graph networks. I remember there are some previous works using graph networks to learn dynamics or using networks to accelerate fluid dynamics. The authors properly cite related works in Section 2.

Quality:
The method looks reasonable to me and the rendering looks good. Just some small suggestions,
1. In the video around 1:30, there are unnatural oscillations. Maybe the collision handling is not perfect.
2. It might be more convenient for readers if some labels can be added to the plots (Fig. 4, 6).
3. It would be better to show the standard deviation in the plot.

Clarity:
Yes.

Significance:
This might be my main concern about this paper.

First, the examples are relatively simple. The topologies are either linear or tree-like. Since graph networks are used, it will be more convincing if the method can simulate more complex graphs, like 2D or 3D nets. I’m also wondering what real-world application can be for this method.

Second, the performance boost is good but not so significant. If I understand the submitted code correctly, the CG solver is implemented using pytorch cuda tensor at an element-wise level. A C++ simulator might be orders of magnitude faster. If the simulator is implemented using C++, I doubt the performance gain of reducing several iterations in CG solver on the cost of running a graph network.

*** After authors’ response

The response points out that the acceleration could be more significant in larger-scale applications, which is good for generalization. The idea itself, using GNN to accelerate the solvers in physics problems, is interesting. I’m glad that the authors provide their code. It is understandable to use python implementation as a proof of concept. More efficient C++ implementation could be future work.


**Time Spent Reviewing:**

4

---

### Official Review · Reviewer_cFjQ · 2021-07-14

**Rating:** 6
**Confidence:** 3

**Summary:**

This paper presents a learning-based technique to speed up the physics simulation of rod structures using PBD. The key motivation is to notice that solving the nonlinear system using CG is the key bottleneck in the physics simulation. To speed up the CG solver, this paper proposes to train a neural network that predicts a good initial guess for CG in the hope that it will reduce the number of iterations before convergence. This paper presents experiments with tens to over a thousand nodes and shows that the speedup technique reduces the number of iterations used in CG by a factor of at most 2.5 (Fig. 6).

**Limitations And Societal Impact:**

See the review above.

**Main Review:**

Overall, this paper makes an attempt to combine learning-based approaches with classical physics simulation. In particular, the paper intentionally avoids training an end-to-end network that fully replaces simulation in a single time step. Instead, it bases its designs on a good understanding/decomposition of the underlying physics simulation technique (PBD) and chooses to train a network that proposes initial guesses to the numerical solver only. The implication is that their method naturally inherits the physical accuracy and stability of PBD because their method technically should lead to almost the same numerical results as the original PBD method. This gives their learning-based model some transparency and good guarantees on its physical correctness compared to an end-to-end network. I like this way of combining a network with a physics engine, and I think the authors should be given credits for this.

My scores cannot be higher for the submission in its current status because I still have some concerns regarding the technical methods and the experiments:

- To me, the improvement from their method over a classic CG seems not significant, and the speedup seems diminished especially when the problem scales up: for the first two examples mentioned in Fig. 4 and Fig. 6, the speedup seems to be between 1 and 3, and for the larger tree example, the speedup is reduced to less than 20%. Is there a reason why the performance degrades when the problem scales up? Even for the complex scene, the matrix size (a few thousand by a few thousand if I understand correctly) is not super large for a modern CG solver, so I wonder if the authors could provide the wall-clock time for simulating one time step for both the small and large examples. If these examples can already be simulated very fast (say in real time), then it might be hard to justify why it is still worth spending tens of hours of training to gain this relatively small speedup.

- Following the comment above, I notice the comparison is against a vanilla CG, which I find quite uncommon in modern physics simulators. Typically, modern simulators will at least equip CG with a preconditioner, which can speed up the vanilla CG at least a few times faster (this is also why I mentioned above that the speedup reported in this paper is not very impressive to me) especially for large-scale systems. There are also more advanced techniques like multigrid solvers that can expedite the vanilla solver even further. All these classical numerical methods do not require extra training time, making it difficult for me to appreciate the speedup obtained in this paper. On a side note, I would be much more excited if this paper proposes to learn a proper preconditioner instead of learning an initial guess to CG because a good preconditioner almost always ensures a decreased number of iterations in CG regardless of the initial guesses.

- I think the discussion of the generalizability of their neural network is very crucial to their method (or any methods trying to combine neural networks with physics engines in general) and I wish it could be elaborated in the main paper. In particular, 1) do I need to retrain the network if I switch to a different example with different topologies (e.g., from rods to trees)? The answer seems to be no from what I understand, but it would be good if the authors can confirm. 2) for the same example, if I change the external forces (e.g., removing gravity or adding wind forces in the tree example) or change to a different boundary condition (e.g., stitching an end of a rod to a wall or ground), do I need to retrain the network or does it still generalize well? 3) if I change the time steps used in physics simulation, do I need to retrain the network or does it still generalize well? 4) is the network described in Fig. 3 agnostic to rigid modes applied to the input? In other words, if I rotate and translate the whole rod system and send it to the network again, would I expect some change in its output?


**Time Spent Reviewing:**

4.5

---

> ### Comment · Reviewer_cFjQ · 2021-08-31
> **Thank you for your rebuttal. My score remains at 6.**
>
> Thank you for your rebuttal letter. I have carefully read your rebuttal, the revised manuscript, and other reviewers' comments, and I maintain my score at 6. However, my feeling is that it is really a borderline paper, and I don't think my major concerns are fully addressed in the rebuttal. Below are my detailed responses:
>
> 1. I am still not convinced that the overall speedup (20% to 50%) is significant, especially because the baseline is a vanilla PBD implementation with a vanilla CG solver. As I suggested in my review before, choosing a good CG preconditioner can boost the speed of a CG solver by a factor of more than 20% to 50%. Furthermore, many advanced methods in graphics can speed up PBD much more, possibly by order of magnitude (e.g., "A Chebyshev Semi-Iterative Approach for Accelerating Projective and Position-based Dynamics" and its related work). While it is beyond the scope of this paper to boost any of the state-of-the-art PBD simulation methods, I still expect a much more significant speedup from boosting a vanilla PBD method with a vanilla CG solver. On a related note, I highly recommend that the authors add the new experiment "Advanced CG" to the manuscript and update the performance data accordingly.
>
> 2. I have some doubts about the importance of a good initial guess in a CG solver. For example, consider an extreme case of solving Ax = b where A is an identity matrix. Regardless of how carefully we choose the initial guess x0, running CG in this system will always converge in 1 iteration. This is also related to my earlier comment that suggesting a good preconditioner would be more exciting than suggesting a good initial guess.
>
> 3. Thank you for your clarification on my questions regarding the generalizability of your method.
>
> My score is still borderline positive mostly because of the other contributions in the paper, e.g., the way it combines a physics engine with a learning-based approach. However, I hope the authors can spend more time and effort looking into the issues I mentioned above.

---

> > ### Author Response · Authors · 2021-09-01
> > **Response to Additional Feedback**
> >
> > Thank you for your additional feedback. We will make sure to carefully address these concerns in the final version. Specifically, we are happy to add the “advanced CG” experiment described in the rebuttal letter as requested under point (1) and provide a clarifying discussion, also w.r.t. other preconditioners.
> > Regarding point (2), we will add a paragraph highlighting the importance of proper initial guesses for CG solvers to balance the discussion. In all experiments that we conducted for the paper we never encountered a case where the initial guess did not matter.
> > We will add the clarifications described in the rebuttal regarding your questions about the generalizability to paper (point (3)).

---

### Official Review · Reviewer_ixu1 · 2021-07-16

**Rating:** 6
**Confidence:** 4

**Summary:**

This paper aims to accelerate the iterative physical simulators using graph neural networks, by combining the traditional iterative rod solver and the graph networks. Instead of an end-to-end way, this paper uses the trained GNN to provide the initial guess of the iterative solver to reduce the number of iterations. In the experiments section, this paper shows the improved run time performance compared with the traditional solver and a KKN method.

**Ethical Concerns:**

I didn’t find ethical concerns in this paper.

**Limitations And Societal Impact:**

The paper properly discusses the limitations that COPINGNet is currently limited to rod simulation using position-based dynamics and mentions the possible future work of applying this method to other iterative methods.

**Main Review:**

#### Strengths
- This paper uses the network output as the initial guess and then performs the iterative solver to correct the position, which is different from other data-driven simulators and the iterative solver can provide better long-term stability compared with the end-to-end methods.
- Compared with the traditional simulation methods, COPINGNet reduces the number of iterations by providing a better initial guess using GNNs which accelerate the simulation.
- This paper gives detailed steps about how to build the graph. And the supplementary video is illustrative.

#### Weaknesses
- To use this method, all the works of a traditional PBD simulator cannot be avoided, and all the physics properties of the physical systems need to be known.

#### Correctness
The technical content of this paper appears to be correct.

#### Clarity
Overall, the exposition of the paper is fairly clear.

#### Relation to prior art
This paper properly discusses the previous work of data-driven physics simulators and graph-based learning.
This work can be seen as a combination of the traditional rod simulator and the GNN-based simulator.
Compared with other GNN simulators, the main difference is that COPINGNet uses the output of GNNs as the initial guess for the iterative solver instead of an end-to-end way. Also, different from many other data-driven methods where the physical system is unknown, this paper assumes all the physics constraints are known to the iterative simulator. And many other GNN-based methods use particles as the graph nodes, while this model uses segments.
Compared with other traditional solvers, this work uses a neural network to provide an initial guess for the solver.

#### Reproducibility
Implementation details and code are provided in the supplementary materials to reproduce this work.

#### Additional Feedback/Questions
- Figure 4 compares COPINGNet’s inference time and the vanilla CG solver’s run time. Is the CG solver also implemented on GPUs? it would be clearer if the devices used to run the COPINGNet and the CG solver are listed since the run-time improvement may also come from the device or implementation difference.
- Is COPINGNet also used for generating the initial guess of the collisionConstraintProjection (line 13 in algo 1)?
- There are other methods that can learn end-to-end simulations using graph networks, is the difference here that the rods simulated in this work are too complicated to get a long-term stable prediction if without using the traditional projection solver to correct the results?
- It might be clearer if the formula of using λ to update p is given in algo 1.
- After training, can the model be used for different time steps or for systems with different physical attributes (e.g., rods with different stiffness)?


**Time Spent Reviewing:**

5

---

### Official Review · Reviewer_W28a · 2021-07-17

**Rating:** 6
**Confidence:** 3

**Summary:**

The authors consider the problem of speeding up a specific form of dynamic simulation - position based dynamics, used as a solver to simulate the dynamic evolution of rods. I believe this is a timely topic, both due to the growing importance of simulations in various communities within AI, and because flexible objects are becoming more relevant in some of these applications.

The core concern of this paper is to find ways to speed up PBD solvers. To the extent that the iterative process of constraint projection is computationally expensive, there is scope for finding surrogates that could perform something like this computation quicker. The author's proposal is to use a Graph Neural Network to act as a surrogate for the analytical calculation. In contract to some other works that aim at replacing the entire simulator, the authors restrict the scope here to only using the neural network to emulating the specific projection step, which then serves as an initialisation, speeding up subsequent computation. This is a good idea and the empirical data supports its use. The empirical experiments in this paper are simulations of free evolution of a rod and a spring, clamped on one side, after an initial perturbation. It is shown that the proposed method can be more computationally efficient than an alternative which only uses the full conjugate gradient optimization step from a zero initialization.

**Ethical Concerns:**

-

**Limitations And Societal Impact:**

The authors explicitly discuss limitations, and their arguments are on point. Ultimately, this is a methodological contribution to solver technology for dynamic simulation, and the limitations pertain to the balance between simulation and learning.

I do not believe this work raises societal issues that need to be considered specifically.

**Main Review:**

I find the work to be sound, and well motivated. There is a need for efficient solvers, and there is also a need for methods that can provide long-term stability, which motivates modifying rather than replacing existing solvers.

The proposed approach - to train a network to emulate the optimization step is a good one. However, as already queried by the reviewers of the previous submission, it is certainly interesting to consider what sort of surrogate is needed and how the resulting method behaves. The authors present a comparison against nearest neighbours, which is a natural baseline. The authors show that while this gives a small boost, the neural network does indeed perform better. Also, the authors note that the network alone would not be a good replacement as it performs very poorly outside the training data regime (which is in accordance with findings in other literature too).

All this raises the question of what can be said about the convergence and error characteristics of such a mixed method, beyond the empirical experiment. To the extent that this is a standard solver with established analysis methods, could one say more about the conditions for long term stability and could that constrain the neural network? Here I have in mind analogies to other numerical methods for speedups of solvers (e.g., the Parareal scheme for time-parallel speedup: https://en.wikipedia.org/wiki/Parareal, which comes with some conditions for stability).

One reason to ask this question is that I am particularly interested in the behaviour of the network when the dynamics become stiff or otherwise numerically difficult. It is not entirely easy to tell from visual inspection of the variations in the scenarios how the ranges of Young's Modulus and rod lengths map to stiffness and longer time scale dynamic evolution - this is where I'd expect the neural network error characteristics to be less favourable. The current implementation requires normalisation of Young's modulus and torsion modulues in order to stabilise the algorithm. I am curious how all this behaves if the system under consideration were such that the underlying dynamics had slow-fast characteristics (perhaps because of inhomogeneous materials like composites or other phenomena like flutter). The paper would be made stronger by including some discussion of these kinds of issues.

Note added post-response:
I thank the authors for their response. I would encourage them to provide the additional information they have offered. Based on the discussion so far (including other reviews and the responses), I maintain my score.


**Time Spent Reviewing:**

2

---

### Author Response · Authors · 2021-08-09
**Rebuttal Letter**

We thank the reviewers for their effort, time, and suggestions. We are happy to read that the reviewers "believe this is a timely topic" with "arguments [...] on point" and “empirical data support[ing] its use” (Reviewer W28a). They appreciated our "fairly clear" contribution which "provide[s] better long-term stability compared [to] end-to-end methods" (Reviewer ixu1) and acknowledged that it "gives learning-based models [...] transparency" (Reviewer cFjQ).

This underlines the high value of our work, and we are convinced that the reviewers’ valuable feedback can be appropriately addressed in the final version.

*** Long-term Stability (Reviewer W28a)
We can further discuss this in the final version. If requested, we could also add a small study on long-term stability of the network part (without the surrounding PBD framework) in stiff cases. We could add rod simulations of different stiffnesses or apply a variation of the moduli along the centerlines representing inhomogeneous material setups. This way, we can quantify numerical stiffness and show concrete thresholds causing failure cases.

*** Need for PBD Solver (Reviewer ixu1)
In contrast to end-to-end learning methods our goal is to improve the stability and speed of existing solvers – we do not want to replace the PBD solver entirely. Understanding the properties of physical systems is a major line of research in various areas. Integrating learned components into existing solvers enables us to leverage the advantages of existing physical solvers. Consequently, our method is an orthogonal line of research compared to end-to-end learning-based methods. As Reviewer cFjQ pointed out, our approach "gives [the] learning-based model some transparency and good guarantees on its physical correctness".

*** Questions (Reviewer ixu1)
(1) The CG solver runs on the CPU and the neural network part on the GPU to get the initial guess, and then solves the equation on the CPU. The main computation cost of both methods come from the CPU ends. The purple line in our figure just shows the ratio of the GPU (the neural network part) end time to the CPU end time of the CG solver, which reduces the speedup from the black line to the red line. We will clarify this in the final version.
(2) No, the neural network is not used for generating the initial guess of the collisionConstraintProjection.
(3) In contrast to end-to-end methods we aim to integrate a learned component into existing physical solvers. We have shown that our approach works for different physical systems, i.e. rod and helix, and branching structures, and it is likely that it generalizes beyond the scenarios we describe in the paper. In fact, we expect that most graph-based physics simulations would show less long-term stability when employing end-to-end methods compared with our work. Exploring integration-based learning for other physical systems is an interesting avenue for future work.
(4) We will add the update formula to Algorithm 1.
(5) The generalization depends on what configurations are provided during training. We did not try different time steps but have shown this for different material parameters (see e.g. Figure 9).

*** Speedup (Reviewer cFjQ)
The reported speedup in Figure 6 is not(!) of a single step of the numerical time integration, but instead a total speedup of the simulation for the respective physical system. For the helix case we report a persistent total speedup of around 50%, whereas for the rod case it is initially over 50% and later stabilizes beyond the training data range at over 20%. Given that the training only has to be done once, this is a significant speed up for simulations that run frequently or for longer periods of time. We are happy to clarify this and provide the wall-clock time.

*** Advanced CG (Reviewer cFjQ)
We have tried a more advanced CG routine by using the result of a simulation on lower resolution geometry (i.e. less particles arranged along the centerline) to initialize the system. Our approach still outperforms this preconditioning strategy, and we are happy to add this.

*** Questions (Reviewer cFjQ)
Yes, we agree that generalizability is crucial when combining neural networks and physics solvers.
(1) We observed performance improvements for collision and complex tree cases although the network was never trained on these discretizations and topologies. In this case (please refer to Section 4, 273–280), it turned out that the performance increase is actually greater on the validation dataset that the model has never been trained on. This indicates that our method is capable of generalizing between different datasets.
(2) Please consider that in lines 259–260 we mention that the collision experiment was performed using the vanilla helix model. In this case, we introduced a boundary condition (ground plane) that the network has never been trained on.
(3) We have not tried different time steps but have shown this for different parameters.
(4) The network can generalize to rigid body transformations as indicated by the inference performance beyond the training time range (Figure 4 and 6).

*** Suggestions (Reviewer 5qbT)
We will add labels, standard deviation, and information on collision detection.

*** Real-World Examples (Reviewer 5qbT)
We consider the tree example to be a complex dynamical system where tree dynamics is coupled in real-time with the fluid dynamics of the wind field. It has been recognized that wind flows in tree crowns exhibit turbulent motion (“Role of buoyant flame dynamics in wildfire spread” by Finney et al. [2015]), which is expected to result in irregular branch swaying. The method for fluid coupling is based on “Interactive wood combustion for botanical tree models”  by Pirk et al. [2017] and is capable of simulating turbulence. This results in complex swaying motions of over 1,000 colliding nodes – we consider this to be a complex use case on par with 2D or 3D nets.

*** Clarification (Reviewer 5qbT)
We would agree that implementing the C++ code would increase the speed of the CG solver. However, in terms of complexity, the neural network part is O(n), and the CG part is at least O(n^2). As the simulation size increases, the inference time of the neural network becomes more trivial compared to the CG solver.

---

### Decision · Program_Chairs · 2021-09-27

**Decision:**

Accept (Poster)

**Comment:**

This paper considers accelerating the simulation of rod dynamics by training a graph neural network to act as an initialisation for an iterative solver. This is an interesting and original paper.
While reviewers were generally in favour of this paper, they had a number of concerns. The primary question, including after the rebuttal, is around the significance of the results. Despite that, there is value in this contribution.
The authors should make the promised changes to the manuscript.